# Nurture Early for Optimal Nutrition (NEON) participatory learning and action women's groups to improve infant feeding and practices in South Asian infants: pilot randomised trial study protocol

Logan Manikam [ID],[1,2] Shereen Allaham [ID],[1] Priyanka Patil,[1,2] Maryan Naman,[2] Zhen Ling Ong,[2] Isabel-Cathérine Demel,[2,3] Neha Batura [ID],[4] Clare Llewellyn,[5] Andrew Hayward,[1] Rajalakshmi Lakshman,[6,7] Jenny Gilmour,[8] Kelley Webb-Martin,[9] Carol Irish,[9] Mfon Archibong,[9] Corinne Clarkson,[10] Daley Delceta,[11] Lily Islam,[12] Seema Bajwa,[13] Sabiha Malek,[13] Jasvir Bhachu,[13] Geromini Pushpakanthan,[13] Michelle Heys,[14,15] Monica Lakhanpaul [ID],[14,16] on behalf of NEON Steering Team

LM and SA are joint first authors. MH and ML are joint senior authors.

For numbered affiliations see end of article.

**Correspondence to**
Dr Logan Manikam;
logan.manikam.10@ucl.ac.uk

## ABSTRACT

**Introduction** Feeding practices developed in early life can impact a child's nutrition, growth, dental health, cognitive development and lifetime risk of chronic diseases. Substantial evidence suggests ethnic health inequalities, and non-recommended complementary infant feeding practices among UK's South Asian (SA) population. Nurture Early for Optimal Nutrition aims to use women's group participatory learning and action (PLA) cycles to optimise infant feeding, care and dental hygiene practices in SA infants <2 years in East London.

**Methods and analysis** A three-arm pilot feasibility cluster randomised controlled trial will assess feasibility, acceptability, costs and explore preliminary effectiveness for proposed primary outcome (ie, reporting on body mass index (BMI) z-score). Multilingual SA community facilitators will deliver the intervention, group PLA Cycle, to mothers/carers in respective ethnic/language groups. 12 wards are randomised to face-to-face PLA, online PLA and usual care arms in 1:1:1 ratio. Primary outcomes are feasibility and process measures (ie, BMI z-score, study records, feedback questionnaires, direct observation of intervention and sustainability) for assessment against Go/Stop criteria. Secondary outcomes are cluster-level and economic outcomes (ie, eating behaviour, parental feeding practices, network diffusion, children development performance, level of dental caries, general practitioner utilisation, costs, staff time). Outcomes are measured at baseline, every 2 weeks during intervention, 14 weeks and at 6 months by blinded outcome assessors where possible. This study will use concurrent mixed-methods evaluation. Quantitative analyses include descriptive summary with 95% CI and sample size calculation for the definitive trial. The intervention effect with CI will be estimated for child BMI z-score. Implementation will be evaluated qualitatively using thematic framework analysis.

## STRENGTHS AND LIMITATIONS OF THIS STUDY

⇒ The three-armed approach will allow for comparisons of online and face to face delivery with usual care.
⇒ The collection and interrogation of process and feasibility data with clearly defined Stop-Go criteria will allow for an informed decision around feasibility of a larger scale evaluation.
⇒ A broad set of secondary outcome measures will allow for estimation of potential impact, in addition to understanding of the feasibility of capturing reliable and complete data measuring these outcomes.
⇒ Project capacity precludes the spacing of group sessions according to religious celebrations which might affect retention to group session.
⇒ Risk of performance and measurement bias as neither participants nor all research staff will be fully blinded.

**Ethics and dissemination** Ethics approval was obtained from University College London (UCL), National Health Service (Health Research Authority (HRA) and Health and Care Research Wales (HRCW)). Results will be published in peer-reviewed journals, presented at scientific conferences/workshops with commissioners, partners and participating communities. Plain language summaries will be disseminated through community groups, websites and social media.

**Trial registration number** IRAS-ID-296259 (ISRCTN10234623).

## INTRODUCTION

The first 1000 days of life is important for growth and brain development. There is

BMJ

mounting evidence that influences during pregnancy and infancy may alter lifetime risk of nutrition and dental related diseases.[1] Feeding practices developed during this period can impact children's nutrition, growth, dental health, cognitive development, and may lead eventually to heart disease, obesity and diabetes.[2]

Britain's ethnically diverse population is mostly disadvantaged across a range of socioeconomic outcomes, forming fundamental causes of ethnic health inequalities in the UK.[3] Some of the widest differences have been observed in the South Asian (SA) population, more so for Pakistani and Bangladeshi communities compared with Indians.[4]

Systematic reviews assessing complementary feeding practices and the sociocultural beliefs underpinning them in children <2 years within SA families in the UK were explored. Despite the adoption of the WHO Infant and Young Children Feeding Guidelines, there remains substantial evidence of non-recommended complementary feeding practices such as early introduction of solids, introduction of minimum dietary diversity or minimum frequency of meals being followed.[5–7] Contributing factors that persisted postmigration included bicultural issues or low acculturation levels and conflicting information among health professionals, extended family, and religious and community leaders.

Effective early life interventions tailored to different ethnic groups have great potential to reduce the development of short-term and long-term conditions and, thereby, lifetime inequalities. At present, however, few of such interventions exist.[8] Traditionally, UK health services provide unidirectional information based on guidelines and National Health Service (NHS) recommendations. Specific ethnic groups may be marginalised by this approach as most advice is not tailored to their cultural practices. There is a need for interventions that target these communities through a top-down unidirectional approach and bring a change originating from within the communities. Therefore, it is important to work in partnership with communities thereby building their capacity to work closely with local authorities/stakeholders. The asset-based community development approach has been successful in community development as this helps communities identify their assets through asset mapping and mobilise them to bring along the desired change.[9]

Based on Lakhanpaul and Panchsheeel's Motivation Awareness Resources Knowledge and Skills model,[10] current practice lacks the space to understand parents' motivation and ability to support families in skills development, all of which is key to providing optimal infant feeding, care and dental hygiene support. Considering the resource constraints of the NHS and its 10-Year Forward Plan which aims to shift the emphasis to the community, highlighting a need for interventions that enable communities to use their available community assets. The participatory learning and action (PLA) cycle is an iterative process led by multilingual facilitators through a four-stage cycle of identifying and prioritising contextual issues, designing problem-solving strategies, implementing these strategies and a postimplementation evaluation. The PLA approach is a low-cost, community-based, culturally sensitive intervention that can be adapted from low-income and middle-income countries (LMICs) to high-income countries, as well as to different population groups and topic areas. This strategy is also recommended by the WHO to improve maternal and infant survival.[11] A meta-analysis of seven cluster randomised controlled trials (RCTs) using PLA in LMICs showed reduced maternal and neonatal mortality by 37% and 23%, respectively.[12]

An RCT and controlled before-and-after study have demonstrated reduced maternal depression, increased exclusive breastfeeding rates and decreased under-5 morbidity.[13] A Mumbai-based RCT also showed improvements in maternal practices and care-seeking behaviour.[14] Recognising WHO recommendations and the success of PLA in LMICs, this approach was adopted for the Nurture Early for Optimal Nutrition (NEON) programme.

In NEON phase 1 (intervention development) was codeveloped with the SA community facilitators (CF) (now community researchers (CRs)/independent observers in NEON phase 2), community members, research team and independent experts (ie, health visitors (HVs), dentists, dieticians, nutritionists, general practitioners (GPs) and midwives). This followed the MRC complex intervention framework, which involved using formative research and prior trials as evidence-based factors; microadaptations involved adjusting language, literacy levels for materials, using picture cards, face-to-face and interactive learning delivery methods. This protocol describes NEON phase 2: pilot feasibility cluster RCT.

## AIMS AND OBJECTIVES

This pilot feasibility study aims to evaluate the feasibility and inform the design and conduct of a definitive cluster RCT comparing NEON women's group PLA cycle versus usual care to optimise infant feeding, care and dental hygiene practices in SA infants aged <2 years in East London.

### Primary objectives

Assess: (1) the feasibility of proceeding to a definitive trial against predetermined Go/Stop criteria (ie, recruitment, retention rates, intervention support, acceptability), (2) intervention fidelity and participant' adherence, (3) implementation of face-to-face versus online intervention arms.

### Secondary objectives

Assess: (1) feasibility, completeness, acceptability and adequacy of blinding when collecting proposed outcomes to establish optimal outcomes and data collection procedures, (2) time needed to competently deliver both versions of NEON intervention, (3) the mean, SD and

intervention effect with 95% CI of the proposed primary outcome, child BMI z-score.

## METHODS AND ANALYSIS

### Design

This three-arm pilot feasibility cluster RCT study is expected to run from July 2021 to May 2023 in three East London boroughs. Eighteen clusters defined as borough wards will be randomised with 1:1:1 allocation to two intervention arms (face-to-face or online NEON women's group PLA cycle) and one control arm (usual care). An online arm enables the trial to continue under COVID-19 restrictions; usual care was selected as control to avoid depriving participants of essential and available infant and maternal care services. Multilingual CFs will deliver interventions in respective ethnic/language groups. Ward-level randomisation is a trade-off between minimising contamination and ensuring maximum SA population representation given the diversity in the ethnic makeup, that is, SA subgroups, of the aforementioned boroughs.

### Patient and public involvement

The research has been codesigned with SA CFs involved in all stages of developing and evaluating the intervention to ensure relevant research questions and an acceptable study design. Development phase, representing different SA communities, is now recruited as CRs to support; protocol development and ethics application by ensuring participants-facing documents are clear, appropriate and sensitively worded; topic guide and questionnaire development; strategy development and troubleshooting (eg, recruitment); interpretation of findings into appropriate and attainable recommendations for practice; review and revision of draft academic papers; dissemination activities and development of plain language summaries.

### Setting

This study will run sequentially in the London Boroughs of Tower Hamlets (TH), Newham (NH) and Waltham Forest (WF), all of which are considered to be among 5% most deprived in England.[15] The boroughs were analysed in terms of ethnic make-up; in each borough, the six wards with the highest density of Asian population were selected based on census data.[16] The most common SA ethnic/language groups will be targeted. Figure 1 shows the number of wards and PLA groups by boroughs. Figure 2 shows the overview of participant flow throughout the trial.

### Randomisation

Randomisation of wards will be done before the recruitment by a separate member at University College London (UCL) using the Research Randomizer software for the 18 wards stratified by borough.[17] The randomised wards will be shared with the RA at the end of recruitment. Once sufficient numbers are recruited in an ethnic/language group, the RA will assign participants to their respective arms. Recruitment will be performed by NHS personnel who will not be aware of the randomisation. CFs/CRs will be made aware of the participant allocation before the PLA sessions start to make contact and invite participants.

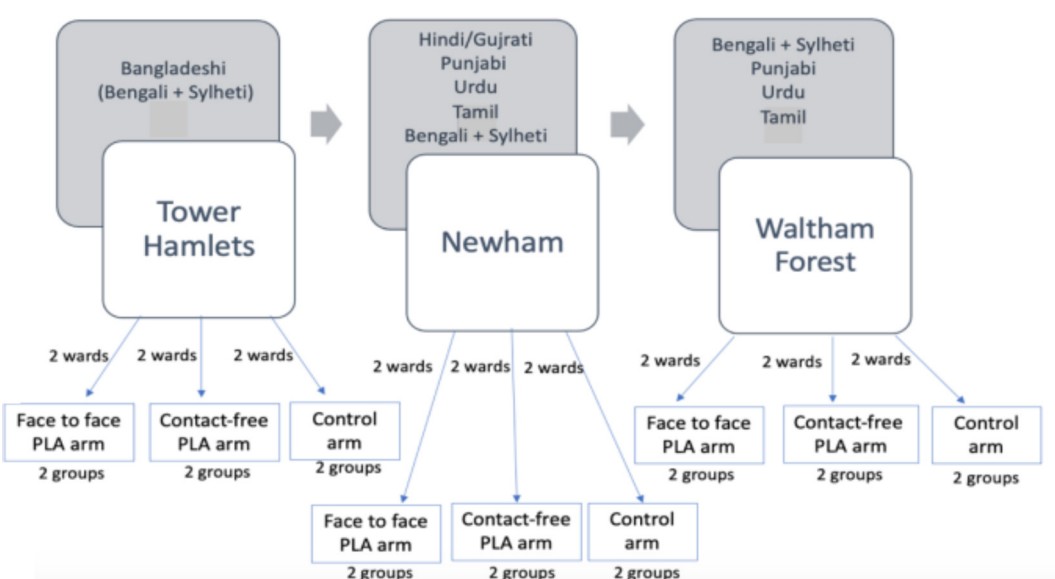

**Figure 1** Overview schematic detailing number of wards and PLA groups by East London boroughs. To illustrate, TH will have two face-to-face Bangladeshi/ Bengali and Sylheti PLA groups (one per ward), two groups of Bangladeshi/ Bengali and Sylheti online PLA group (one per ward) and two Bangladeshi/ Bengali and Sylheti control groups (one per ward). The more the ethnic/ language group in the borough, the more the intervention and control groups. PLA, participatory learning and action.

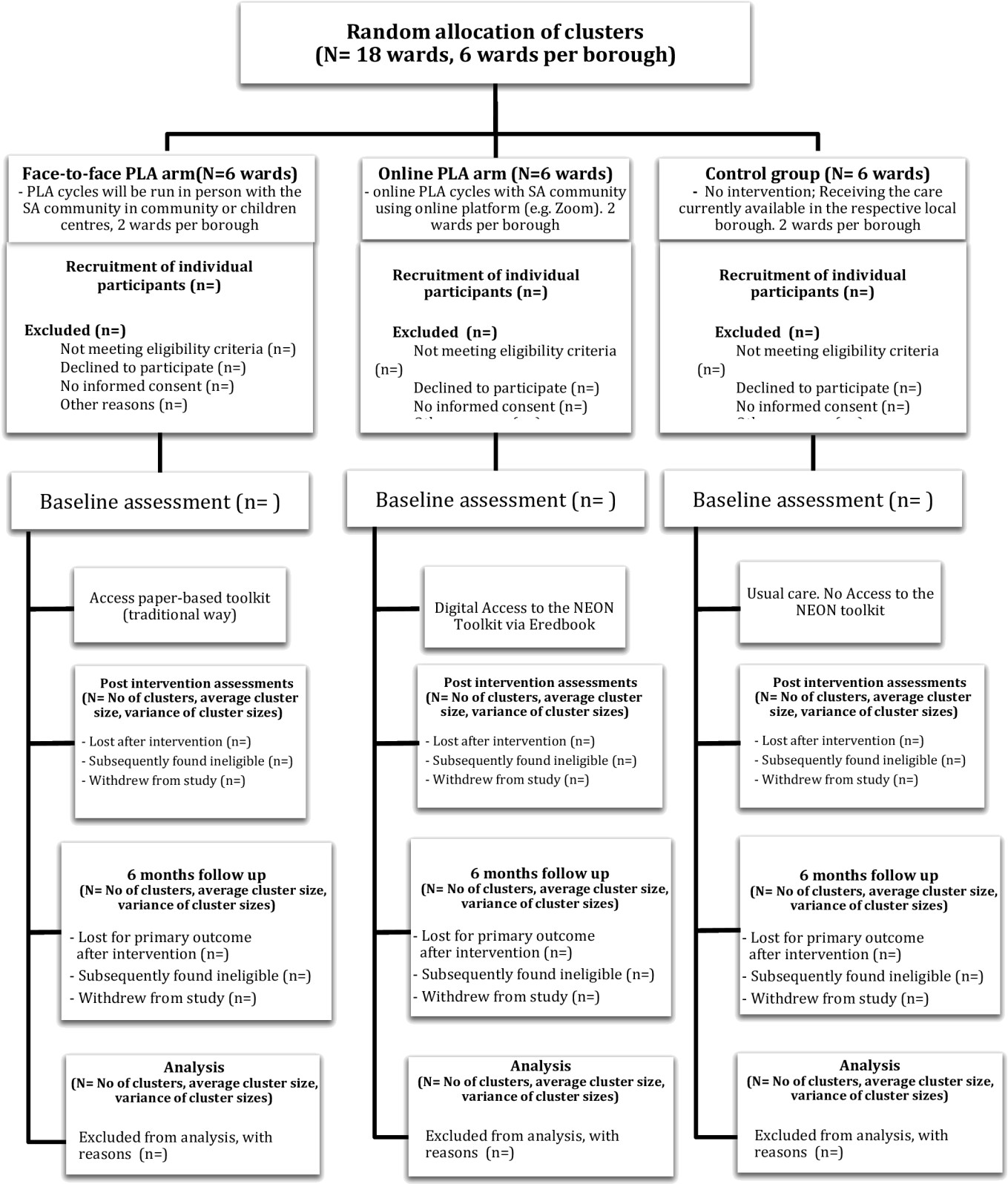

**Figure 2** Overview of the NEON pilot feasibility RCT design and participant flow. NEON, Participatory Learning and Action; PLA, participatory learning and action; RCT, randomised controlled trial; SA, South Asian.

## Eligibility criteria

### Participants

Inclusion

► Mothers or female carers of an infant <24 months.

► Indian, Pakistani, Sri Lankan or Bangladeshi background.

► Resident in a randomised study ward in TH, NH and WF.

► Willing and able to provide informed consent.

Exclusion

► <18 years old.
► Anticipating moving out of a priori defined geographical areas before or after delivery.
► Current or recent participation in another study within 4 weeks of trial commencement.

### Staff (CF) eligibility

► Female.
► Have at least one child, preferably <24 months.
► From the SA community in TH, NH or WF.
► Able to read and write.
► Fluent in English and another local language (Hindi, Bengali, Sylheti, Urdu or Tamil).
► Understand SA social norms, values and culture within study boroughs.
► Known and respected by their local community.
► Motivated to address issues related to infant growth and development.
► Able to manage a group and have some leadership qualities.

### Sample size

With communication and cultural sensitivity as key drivers of the NEON intervention, the sample size will partly depend on the coverage of ethnic and language groupings. The choice of ethnic/language groups were informed by the 2011 ONS Census[16] and input from CFs stressing that, while language is important for communication, people's practices are embedded in different ethnicities.

The main ethnic/language group(s) identified are: one group in TH (Bangladeshi/Bengali and Sylleti); four groups in NH (Indian/Gujrati, Indian/Punjabi, Pakistani/Urdu and Sri Lankan/Tamil); three groups in WF (Indian/Punjabi, Pakistani/Urdu, Sri Lankan/Tamil). In each borough, one women's group PLA cycle will run per ethnic/language group per ward.

We aim to run 20–32 women's group PLA cycle (with equal numbers of face-to-face and online PLA groups) at 6–8 participants per PLA group. Including control groups totals to 288–384 participants. Assuming an 80% recruitment or retention rate, this enables us to be 85% confident that the true population rate will fall between 0.76% and 0.84% (precision=0.04). This was estimated using a geometric mean cluster size=16 participants, cluster size coefficient of variation =0.7 and intracluster correlation coefficient (ICC)=0.02.

### Recruitment

#### Participants

Due to the diversity of the target population, multiple recruitment methods are needed to maximise reach and minimise inequalities or non-representative samples.[18 19] For instance, relying solely on online recruitment may be ineffective due to digital poverty. Thus, other mediums including leveraging on existing social networks (snowballing), identifying and fostering collaborations with community leaders, and creating clear and succinct recruitment materials[20] were employed. All study promotional materials (eg, participants information sheets, posters) will be available in English and local languages. To standardise the recruitment process, we will provide the same recruitment script and study materials to all recruiters. Potential participants will contact the RA, who will confirm eligibility, consent and register them into the study.

Participant recruitment will follow three strategies with regular advice by CRs to maximise reach:

(1) Recruitment through CFs: CFs will share study materials with their network to invite eligible participants with subsequent snowballing. We will provide training to standardise CF recruitment. (2) Recruitment through HVs, GPs and Midwifery teams: HVs, GPs and midwives who have been identified based on their geographical proximity to study wards will share study materials to eligible participants. Details of those interested will be given to the RA. (3) Online recruitment: Online social media campaigns on Facebook and Instagram will be used. See online supplemental material 1 for advertisement materials. Recruitment bias will be limited due to the above-mentioned multiple channels of recruitment and all recruiters will be blinded thereby not knowing the allocation details at the time of recruitment.

### Staff

We will recruit CFs from our local stakeholders' network. Recruitment via CFs is particularly useful for studies with underserved or traditionally hard-to-reach populations.[21] CFs are leaders/champions in their local community who, with training and support, help improve the health and well-being of their communities. In line with NIHR's 'INVOLVE' guideline, CFs will be paid £30 for each PLA meeting and additional £5 to cover travel costs for face-to-face meetings. A comprehensive manual for the CFs will be developed with PLA experts. A tool kit will be provided to support the learnings. The CFs will attend a 3-day comprehensive workshop and additional sessions will be conducted in between the eight-meeting cycle as a refresher and address any challenges. Biweekly meetings with the RA will be conducted to discuss learnings from the sessions. The CFs will be paid through vouchers for their time similar to a 0-hour contract. HV, GP and midwives would be identified and recruited by our study partners at each study ward. Payments will be supported by NHS support costs and decided with the relevant clinical lead.

### Allocation concealment and blinding

First, treatment allocation will be concealed from participants and recruiters during participant recruitment to reduce recruitment bias. After recruitment has been completed for that ethnic/language group in the borough, the RA will reveal ward allocation to

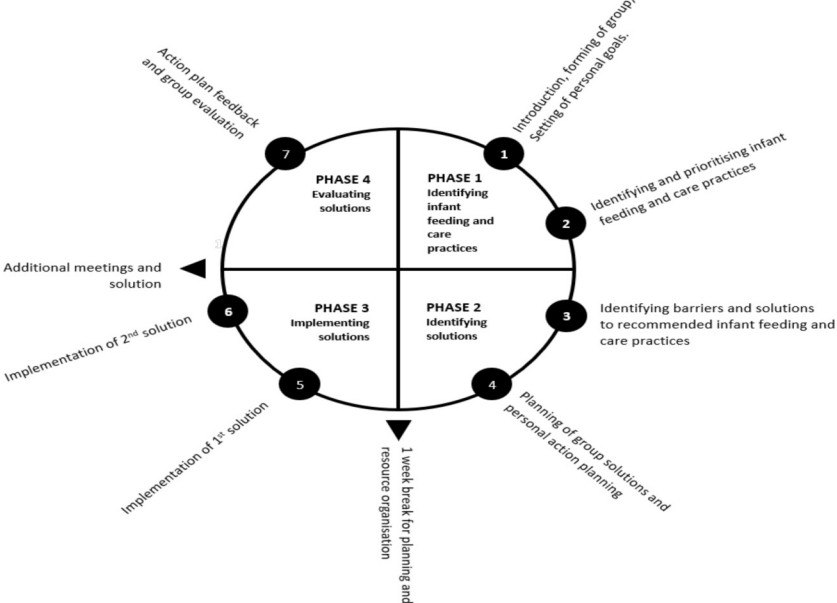

**Figure 3** NEON Women's Group PLA Cycle. NEON, Participatory Learning and Action; PLA, participatory learning and action.

participants and CFs via text or email, listing postcodes by trial arms to allow CFs to effectively organise and deliver the intervention.

Second, outcome assessors (CRs) will be blinded where possible to reduce measurement bias. This will be achieved by CRs logging the data collected onto an electronic platform that automatically links participant ID to maintain blinding. Baseline measurement of BMI will be done by HVs in each borough who will be blinded.

### Intervention
#### NEON women's group PLA cycle
The co-adapted NEON women's group PLA cycle tailored to ethnic/language groups will be delivered face-to-face at community/children's centre or online over 14 weeks (1 session per 2 weeks). HVs, GP and midwives' teams will be invited to 1–2 sessions to provide evidence-based information on participants' request. To reduce study contamination, participants may not switch wards (ie, switch arms) after treatment allocation is revealed. Figure 3 illustrates the aim of each PLA session.

A culturally sensitive intervention package has been codeveloped consisting of:
1. PLA group facilitator manual.
2. Picture cards with recommended and non-recommended infant feeding, care and dental hygiene practices, and facilitators/barriers to uptake.
3. Healthy baby food recipes.
4. Community asset maps (eg, low-cost fruit/vegetable shops).
5. List of relevant resources and services.

Online arm participants may access this toolkit through the eRedbook platform and receive NEON information the via the app or website. A logic model

(online supplemental material 2) shows the relationship between these activities and resources and their intended short-term, medium-term and long-term effects.

#### Control (usual care)
In all wards, HV teams have regular mandatory postnatal visits for all families of newborns and infants. These visits are immediately after birth, 6–8 weeks, 12–16 weeks, 1 year and 2–2.5 years. One optional prenatal visit is conducted by HVs between 28 and 32 weeks of pregnancy. Online supplemental material 3 lists other initiatives and resources available in each study borough as part of usual care.

### Outcome
#### Primary outcomes
*Feasibility and process outcome measures*
We will collect quantitative and qualitative data to assess Go/Stop criteria (table 1) on recruitment and retention rates, intervention support, acceptability, fidelity and participant adherence using:
1. Study record: An enrolment log will record all eligible participants, total enrolled, reasons for non-participation, number followed up on child BMI z-score, and the date on how many people responded to adverts/invitations. Prescreen failure logs will record those eligible but not enrolled, with reasons. PLA session audio/video recordings will be collected using secured Dictaphone, video cameras, CCTV and/or Zoom recording.
2. Process measurements: These include Participants' Feedback Questionnaire, Facilitator's Feedback Form, PLA Cycle Meeting Register, direct observation of intervention delivery and CF performance

**Table 1** Go/Stop criteria for the definitive trial*

| Definite Go | Definite Stop |
|---|---|
| ≥50% of eligible participants consenting to pilot feasibility study. | <40% of eligible participants consenting to pilot feasibility trial. |
| ≥80% of mothers attend ≥60% of planned sessions in the intervention arm. | <20% of mothers attend ≥60% of sessions as planned in each intervention arm. |
| Retention of ≥70% of consented participants for primary outcome data collection. | Retention of <50% of participants at 6 months. |
| High intervention support with respect to content, frequency, duration and quality. | Low support of intervention procedures. |
| Intervention is perceived as acceptable. | Intervention perceived as unacceptable. |

*The study will have to meet the Definite Go criteria in order for the study to be feasible and to be able to proceed. However, if any one of the Definite Stop criteria is met then the study stops. The 'Go/Stop' indicates that it is either feasible or not feasible to proceed to a definitive trial. The progression rules have been approved by the steering and data management team. Should any of the progression rules not be met, assessments and adjustments of the NEON pilot feasibility RCT will be negotiated before proceeding to a definitive trial.
NEON, Nurture Early for Optimal Nutrition; RCT, randomised controlled trial.

by CRs, and Sustainability Assessment Tool (online supplemental material 4).

### Secondary outcomes
#### Proposed outcomes for the definitive trial
All other individual-level, cluster-level and economic outcome measures proposed for the definitive trial will be assessed on the response, completion rates, acceptability and the adequacy of blinding for outcome assessors during data collection. Table 2 summarises outcome measures and data collection timing. See online supplemental material 5 for collection tools.

### Primary individual-level outcome
1. Child BMI z-score: HVs will measure infants' length/ height, head circumference and weight. The RA will refer to the WHO 2006 growth standard,[2] using age-adjusted and gender-adjusted length or height(m)/ weight (kg)$^2$ to provide a comparison on the cluster level.

### Secondary individual-level outcomes
1. Children's feeding behaviour will be measured using six of eight domains adapted from the validated Children's Eating Behaviour Questionnaire deemed suitable for infants: food responsiveness; enjoyment of food; emotional overeating; desire to drink; satiety responsiveness; slowness in eating; emotional undereating; and food fussiness.[7 8 22] Obesity risk could

be linked with external eating, emotional overeating and food fussiness.[7]
2. Parental feeding style will be assessed with a validated self-reported Parental Feeding Style Questionnaire with four scales: emotional feeding; instrumental feeding; encouragement to eat and control overeating.[23]
3. Audio/video recording of child eating behaviour and parental feeding practices[24] involves asking participants about the meal (eg, name, ingredients) and creating annotations that link to behaviour codes identified from NEON formative study.[25] Before-and-after audio/video recordings will be thematically analysed using Elan Software[26] to measure intervention effect.
4. Child food intake: The 4-day food diary[7] is a self-reported, prospective, open-ended survey that collects detailed quantitative estimates of food consumption of infants between 4 and 18 months.[6] It assesses nutrient intake (eg, total fat, total carbohydrate, salt, sugar) by comparing with age-specific/sex-specific UK dietary reference values.[27] Dietary data will be presented by age group (eg, 4–6 months, 7–9 months, 12–18 months).
5. Network diffusion of study materials between participants and their communities: CRs will track the number of downloads using the eRedbook platform. Participants will also complete a questionnaire on the number of people they shared the material with, their family size, age, gender, relationship to the participant and platform used.
6. Equality impact assessment (EIA): This trial should reflect the diversity of the target population by enabling equal access while accommodating different needs. An EIA tool will systematically assess likely effects of the intervention on people (eg, with respect to disability, gender, ethnicity, age, sexual orientation, religion/belief), mitigate adverse impacts and identify active steps to address existing disadvantage and promote equality.

### Secondary cluster-level routine outcomes
Although this intervention acts on the individual-level, we expect a social diffusion of good infant feeding and dental hygiene awareness and practices in the medium term to long term (online supplemental material 2: logic model). While immediate cluster-level changes are not expected, we would like to pilot cluster-level routine data collection.
1. Development performance of children: The Ages and Stages questionnaire (ASQ-3) includes communication, gross motor, fine motor, problem-solving and personal-social domains.[5] Three versions are available: 24 months, 27 months and 30 months. ASQ-3 data collected by HVs during mandated visits are stored in EMIS/RIO databases.
2. GP healthcare utilisation data is available in participants' medical record.
3. Level of dental caries and other signs of tooth decay is available in NHS dental services records for registered patients beginning at 6 months of age and

**Table 2** Schedule of outcome assessments

| Outcome | Outcome measures | Data collection | Baseline | Every 2 weeks (end of each PLA meeting) | 14 weeks (end of PLA cycle) | 6 months |
|---|---|---|---|---|---|---|
| | | | | Timing | | |
| Proposed primary individual-level outcome | Individual Child BMI z-score | RA, CRs | ✓ | | ✓ | ✓ |
| Proposed secondary individual-level outcome | Children feeding behaviour* | RA, CRs | ✓ | | ✓ | ✓ |
| | Parental feeding style* | RA, CRs | ✓ | | ✓ | ✓ |
| | Audio/video recording of child eating behaviours and parental feeding practices | RA | ✓ | | ✓ | ✓ |
| | 4-day food diary† | RA, CRs | ✓ | | ✓ | ✓ |
| | Network diffusion†‡ | RA, CRs | ✓ | | | ✓ |
| Proposed secondary cluster-level outcome | Children's development performance | RA, CRs | | ✓ | | ✓ |
| | Level of dental caries | RA | ✓ | | | ✓ |
| | GP healthcare utilisation | RA | ✓ | | | ✓ |
| Economic outcome | Cost tool | RA | ✓ | | | ✓ |
| | Partners time questionnaire | RA | | | | ✓ |
| Process outcome‡ | Participants feedback* | RA, CRs | | ✓ | | |
| | PLA cycle meeting register* | RA, CRs | | ✓ | | |
| | Direct observation | RA, CRs | | ✓ | | |
| | Sustainability assessment | RA, CRs | | ✓ | | |

*The study will have to meet the Definite Go criteria in order for the study to be feasible and to be able to proceed. However, if any one of the Definite Stop criteria is met then the study stops. The 'Go/Stop' indicates that it is either feasible or not feasible to proceed to a definitive trial. The progression rules have been approved by the steering and data management team. Should any of the progression rules not be met, assessments and adjustments of the NEON pilot feasibility RCT will be negotiated before proceeding to a definitive trial.

†Community Facilitators will assist in administering or completing these data collection tools before sending them to RA/CRs during the intervention period.

‡For intervention groups only.

BMI, body mass index; CR, community researchers; GP, general practitioner; NEON, Nurture Early for Optimal Nutrition; PLA, Participatory Learning and Action; RA, research assistant; RCT, randomised controlled trial.

approximately every half-year following initial visit.[17 28 29]

### Economic outcomes
1. Cost: The RA will use a spreadsheet will capture the cost of the NEON Intervention Development and pilot feasibility RCT phases every 6 months.
2. Partners' time used: A questionnaire will capture time spent for the delivery of NEON-associated programmes.

### Data collection and management
Outcome data will be collected at baseline, every 2 weeks during intervention (for process measures), 14 weeks immediately postintervention, and 6 months follow-up (for economic data). During the intervention period, several outcome measurements will be administered by CFs to participants or completed by CFs and sent to CRs/RA. Other outcomes and postintervention outcomes will be assessed directly by CRs/RA (table 2). Each case will be assigned an ID. Data collected digitally or in paper format will be stored in UCL S: Drive and/or UCL cabinets. We will use RedCap electronic data management system[30]; the RedCap API allows flexible and straightforward data import and export to data analysis software.

We will form a data sharing agreement with study partners for the RA to extract and link routine cluster-level data to trial participants. To guarantee confidentiality, pseudonymised participant data will be stored on encrypted, password-protected computer or UCL S:Drive accessible only to study staff and authorised personnel. Password-protected personal data will be stored separately from trial data on secure UCL computers. Only pseudonymised quotes or data from audio/video recording may be published. Personal data and audio/video recordings will be stored for 3 years poststudy and destroyed; research data will be stored for 20 years poststudy.

### Analysis
Overall, quantitative and qualitative data will be analysed concurrently at multiple time points to identify early

problems that are rectified as the trial progresses. Quantitative data such as the BMI z-score will be descriptively summarised using mean and SD for continuous variables, number with percentages for categorical variables and 95% CI with a breakdown of participants by trial arm, ethnic/language group and borough where relevant. An intention-to-treat analysis will also be considered. Results will be collectively assessed against the Go/Stop criteria. Before progression to the definitive trial, we will systematically and rigorously assess areas of improvement using a structured discussion tool: a process for decision-making after pilot and feasibility Trials.[31] All statistical analyses will be done by an independent statistician who would be blinded.

### Sample characteristics
Participant demographic characteristics will be descriptively summarised by trial arm. We will check for recruitment bias across different arms at baseline and compare later recruits who are prone to unblinded recruitment.

### Feasibility and process measures
Across the three arms, we will check for any differential recruitment or retention rates and any consequent systematic differences in participant characteristics over time, as control arm participants may be less keen to participate or remain in the study. For the two intervention arms, we will conduct a mixed-methods implementation evaluation to explore the possibility of a blended approach in the definitive trial as some features of each delivery mode might be particularly salient. Qualitative data will be analysed thematically using framework analysis that include participants' motivation for engagement, expectations, experiences, intervention acceptability, implementation barriers and suggestions for improvement. Negative 'deviant' cases will inform interpretation.

### Outcome measures for the definitive trial
1. Feasibility of collecting outcome measures: To assess completeness, degrees of missingness at the individual item level and the entire outcome measure will be reported overall, by time point and trial arms. Other feasibility measures would be captured by analyses of process measures above.
2. Intervention effect for proposed primary outcome (BMI z-score): Conditioned on low recruitment bias and sufficient adherence to trial protocol, we will estimate the difference in BMI z-score between each intervention arm vs the control arm. We will consider methods to pool data across age groups depending on the actual sample age breakdown.
3. Sample size and power calculation for the definitive trial: Data from the pilot feasibility trial will enable us to determine (1) the smallest difference of clinical importance (through multiple testing of several key endpoints for the proposed primary outcome), (2) the clinically justifiable power-significance level or scientifically acceptable probability of 'false positive' (type I error) results and (3i) whether we need to adjust the calculated sample size for expected level of non-compliance/drop-in and drop-outs. We will report the ICC.
4. Economic Evaluation: This will be conducted from a provider and user perspective which includes direct cost (eg, time, resources spent on training and participation of CF, participants' overall household consumption and childcare costs). We will estimate the total cost, average annual cost, cost-effectiveness, equity impact analysis and compare across three arms.

### Monitoring
The principal investigator (PI) will be responsible for day-to-day monitoring and study management.

The NEON Steering and Data Management Team will ensure adherence to all relevant regulations and principles of good clinical practice. The team consists of multidisciplinary experts in child health, practitioners, PLA expert, SA independent observers from NEON 1 and Queen Mary University of London Pragmatic Clinical Trials Unit. Committee meetings will be held every 3 months to review the Go/Stop criteria (table 1) and interim analyses. The team will agree to the trial protocol and facilitate any necessary protocol amendments; provide independent expert advice (including on project management governance); monitor progress; ensure adequate deadlines are set and met; monitor and advise NEON Core Team (Trial Management Group) on strategic decisions in light of new evidence; and ensure successful delivery by meeting and reporting on study progress with the Core Team biannually. The PI and other Core Team members will attend all necessary meetings and report on the study progress.

Any adverse events or safety concerns will be captured in the Participant Feedback Questionnaire and Facilitator Report Form (every 2 weeks during intervention period) and reported to the steering and safety monitoring team.

The UCLH/UCL Joint Research Office, on behalf of the Sponsor (UCL), may conduct random audits in accordance with the UK Policy Framework for Health and Social Care Research and UCL's policies and procedures.

### Ethics and dissemination
Ethical approval has been obtained from UCL Research Ethics Committee and NHS Health Research Authority. The participant information sheet will be provided in English or participants' native language in-writing and/or verbally by CFs for informed consent. The RA or CRs will obtain written or audio-recorded consent ≥24 hours later.[32] Participation will be voluntary and they may withdraw anytime without prejudicing treatment or usual care. No incentives will be provided except for small reimbursements (eg, childcare at the local centre where PLA meetings take place).

UCL will provide insurance cover and we will also signpost any vulnerable participants who require protection from harm to relevant safeguarding organisations.

Regular progress updates will be shared via UCL or NIHR websites and patient group newsletters. Study findings will be submitted for 3–5 publications in high impact factor peer-reviewed journals and presented at national and international conferences. Researchers and community members from the steering team will contribute to confirming analysis and write-up, adhering to authorship guidelines. To reach wider audiences, we will (1) disseminate plain language summaries for each academic paper through community groups, social media and You Tube; (2) present findings at lay-person meetings at community or children centres through the NIHR CLAHRC network and (3) organise 2–3 annual workshops with commissioners, partner organisations and NEON CFs to share key findings and recommendations.

**Author affiliations**
[1]Department of Epidemiology and Public Health, UCL Institute of Epidemiology and Health Care, London, UK
[2]Aceso Global Health Consultants Pte Limited, Singapore
[3]Faculty of Life Sciences and Medicine, King's College London, Guy's, King's and St. Thomas' School of Medicine, London, UK
[4]Institute for Global Health, University College London, London, UK
[5]Department of Behavioural Science and Health, UCL Institute of Epidemiology and Health Care, London, UK
[6]Public Health Directorate, Cambridgeshire County Council, Cambridge, UK
[7]Medical Research Council Epidemiology Unit, University of Cambridge, Cambridge, UK
[8]Tower Hamlets GP Care Group, Mile End Hospital, London, UK
[9]Children's Health 0-19 Service, London Borough of Newham, London, UK
[10]Public Health Directorate, London Borough of Waltham Forest, London, UK
[11]Chingford Health Centre Waltham Forest, London, UK
[12]London Borough of Tower Hamlets, London, UK
[13]London Borough of Newham, London, UK
[14]Department of Population, Policy & Practice, UCL Great Ormond Street Institute of Child Health, London, UK
[15]Specialist Children and Young People's Services, East London NHS Foundation Trust, London, UK
[16]Whittington Health NHS Trust, London, UK

**Acknowledgements** The authors would like to acknowledge the contribution of the NEON Core Team, Steering Team and all health experts who contributed to this study. We would like to thank the Women & Children First Charity and First Steps Nutrition Trust for their valuable contributions and guidance throughout the study. Steering team members had an opportunity to critically review results and contribute to the process of finalising this paper. The authors would like to thank the National Institute of Health Research (NIHR) Academy and the NIHR Collaboration for Leadership in Applied Health Research and Care North Thames for funding the NEON study. This work is also supported by the NIHR GOSH BRC. The views expressed are those of the author(s) and not necessarily those of the NHS, the NIHR or the Department of Health.

**Collaborators** In addition to the authors, members of the NEON steering team consist of Prof Atul Singhal, Prof Mitch Blair, Joanna Drazdzewska, Dr Sonia Ahmed, Amelie Gonguet, Gary Wooten, Dr Ian Warwick, Vaikuntanath Kakarla, Phoebe Kalungi, Prof Richard Watt, Prof Audrey Prost, Dr Edward Fottrell, Ashlee Teakle, Prof Oyinlola Oyebode, Keri McCrickerd, Dr Rana Conway, Professor Lisa Dikomitis, Mari Toomse-Smith, Scott Elliot, Julia Thomas, Aeilish Geldenhuys, Chris Gedge, Kristin Bash, Dr Dianna Smith, Kate Questa, Dr Megan Blake, Prof Gary Tse, Dr Queenie Law Pui Sze, Gavin Talbot, Dr Chiong Yee Keow, Dr Angela Trude, Prof Lindsay Forbes, Dr Nazanin Zand, Lakmini Shah, Subarna Chakraborty, Yeqing Zhang, Sumire Fujita, Dina Mobashir, Natasha Chug, Tala El Khatib and Delaney Douglas-Hiley.

**Contributors** ML, LM, MH, NB, CL, AH and SA conceived the original concept of the study and designed the research methodology. SA carried out the intervention development meetings, workshops, codevelopment of the intervention toolkit, wrote and devised the paper along with LI, SB, SM, JB and GP. LM, ML, SA, PP, MN, ZLO, NB, CL, AH, RL, JG, KW-M, CI, MA, CC and DD validated the study and revised the manuscript critically for important intellectual content. I-CD was involved in drafting materials for the NEON meetings and workshops of the intervention development phase. MN and I-CD contributed to the manuscript writing, and prepared it for submission. LM, SA and ML had primary responsibility for the final content. All authors read and contributed to reviewing the study data, the designing of the manuscript, and the approval of the final manuscript.

**Funding** LM, SA and PP are funded via a National Institute for Health Research (NIHR) Advanced Fellowship (Ref: NIHR300020) to undertake the Pilot Feasibility Cluster Randomised Controlled Trial of the NEON programme in East London. ML was funded by the NIHR Collaboration for Leadership in Applied Health Research and Care (CLAHRC) North Thames.

**Disclaimer** The views expressed in the publication are those of the author(s) and not necessarily those of the sponsor (UCL), funder (NIHR), study partners (Tower Hamlets GP Care Group, London Borough of Newham Council, Waltham Forest Council).

**Competing interests** None declared.

**Patient and public involvement** Patients and/or the public were involved in the design, or conduct, or reporting, or dissemination plans of this research. Refer to the Methods section for further details.

**Patient consent for publication** Consent obtained directly from patient(s).

**Provenance and peer review** Not commissioned; externally peer reviewed.

**ORCID iDs**
Logan Manikam http://orcid.org/0000-0001-5288-3325
Shereen Allaham http://orcid.org/0000-0003-0275-3228
Neha Batura http://orcid.org/0000-0002-8175-8125
Monica Lakhanpaul http://orcid.org/0000-0002-9855-2043

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
