## [Reviewer comments · BMJ Open]

ARTICLE DETAILS

TITLE (PROVISIONAL)	Nurture Early for Optimal Nutrition (NEON) participatory learning and action women's groups to improve infant feeding and practices in South Asian infants: pilot randomised trial study protocol
AUTHORS	Manikam, Logan; Allaham, Shereen; Patil, Priyanka; Naman, Maryan; Ong, Zhen Ling; Demel, Isabel-Cathérine; Batura, Neha; Llewellyn, Clare; Hayward, Andrew; Lakshman, Rajalakshmi; Gilmour, Jenny; Webb-Martin, Kelley; Irish, Carol; Archibong, Mfon; Clarkson, Corinne; Delceta, Daley; Islam, Lily; Bajwa, Seema; Malek, Sabiha; Bhachu, Jasvir; Pushpakanthan, Geromini; Heys, Michelle; Lakhanpaul, Monica

VERSION 1 – REVIEW

REVIEWER	Hettiarachchi, Manjula University of Ruhuna, Faculty of Medicine
REVIEW RETURNED	06-Jul-2022

GENERAL COMMENTS	This is the study protocol, hence no comments to the study team.
--

REVIEWER	Mehta, Kala University of California, San Francisco
REVIEW RETURNED	12-Jul-2022

GENERAL COMMENTS	This is a well written study protocol on a 3 arm pilot trial regarding complementary feeding practices in south asian infants residing in the United Kingdom. A few items deserve comment for this overall well thought out study. General comments For this pilot feasibility rct, no specific feasibility aims are mentioned. Unclear what noncomplementary feeding practices are targeted? Why does the trial need 3 arms? Is this perhaps overcomplicated? Specific comments p 6 lines 12-17, what are the disparities faced? Why not be clear and state numbers, direction p 6 lines 27-40, this paragraph is written as if there is a deficit in what the south asian community members are doing. Please consider reframing using an assets based framework p 7 line 27-31, little attention is paid on whether practices learned may be sustained beyond the intervention. How will this be measured/quantified p 7 line 31, many more details on the 'online' arm are needed. Who will access the information? will it be acceptable if a husband brother has the computer or cell phone?
---

	p 8, what was codesigned? How do you think the co design influenced the intervention method p 8 lines 41-44 need more details on the randomization and the way randomization would be concealed p 9 lines 11-24, how will CFs be paid? trained? if trained, by whom? what topics? p 10, lines 20-35, recruitment through CBOs? Will information be retained on how the participant was recruited? Could the method of recruitment bias the sample? p 11, line 58 what does " CF unblinded debrief mean?" In summary, if these points are considered by the authors, I believe this study protocol will make a nice contribution to the field. Thank you for the opportunity to comment.
--	---

REVIEWER	Kerr, Jessica University of Otago Christchurch
REVIEW RETURNED	21-Oct-2022

GENERAL COMMENTS	Abstract & Intro  • NEON in full. • Sometimes there are no spaces in between the brackets, please add the spaces where appropriate. • Typo page 4, line 16 • The primary and secondary outcomes are unclear, which is BMI z-score? I think BMI is the proposed primary for the full trial, but not for the pilot? Please make that clear in abstract. • Unclear sentence page 6, line 38 • I felt like there was too many acronyms to keep track of and it made reading and following difficult. Methods  • Because it's expected to run from July 2021, it was unclear what stage this was at? Perhaps the authors could now update with more accurate dates or estimates? • I also wondered if the protocol may need to be updated to clarify how the intervention was impacted by covid restrictions and how these intricacies will be handled in roll out and analysis etc • Again, I think there are too many acronyms and abbreviations, I don't think these are all needed. • What is the rationale for excluding mothers < 18 years? • Please revise tense throughout methods, sometimes these swap between past, present and future tense. • What training will facilitators receive to ensure this is standardised? • Outcome section table is nice, but perhaps confusing about which are the primary outcomes for pilot vs. full trial and as this paper is a protocol for the pilot, not the trial. Perhaps some colour coding or an extra column could be used to clarify this. • What is the rationale for only 1 of the outcomes measured at 12 months • Outcome table – please clarify if the 6m and 12m measures are after baseline or after study end? Partner's time? Partner of who • The subheadings used in outcomes section are at odds with those listed in the abstract, please make sure it is always clear what the primary/secondary outcomes are for the pilot vs. proposed full trial. • Will BMI z-score be converted to determine overweight/obesity, or only retained as continuous measure?
--

	Analysis  • As analysis will be done during the intervention, who will do this and will they be/remain blinded? • Will intention to treat analysis be considered? • Will there be sensitivity analysis for facilitators or assessors unblinded during the intervention/measurement periods? References Could do with some reformatting and checking
--	---

VERSION 1 – AUTHOR RESPONSE

Reviewer: 1

Dr. Manjula Hettiarachchi, University of Ruhuna, University of Limerick

Comments to the Author:

This is the study protocol, hence no comments to the study team

Reviewer: 2

Dr. Kala Mehta, University of California, San Francisco

Comments to the Author:

This is a well written study protocol on a 3 arm pilot trial regarding complementary feeding practices in south asian infants residing in the United Kingdom. A few items deserve comment for this overall well thought out study.

General comments

For this pilot feasibility rct, no specific feasibility aims are mentioned.

Thank you, this is referred within text

The specific feasibility aims for this study are assessing the feasibility and acceptability of the definitive trial intervention which includes the trial design and procedures, and aid in sample-size calculations for the definitive trial.

- Feasibility to recruit participants and implement the intervention
- Acceptability of the intervention to participants and community facilitators (CFs)
- To assess the capabilities of the CFs to deliver the intervention with the training provided
- Suitability of measuring the outcome measures

Unclear what noncomplementary feeding practices are targeted?

Untimely introduction of complementary foods as early as two months of age, not achieving the minimum dietary diversity or minimum meal frequency as recommended by the WHO.

Text within Manuscript:

Despite the adoption of the WHO Infant and Young Children Feeding Guidelines, there remains substantial evidence of non-recommended complementary feeding practices being followed [5-7]. from referenced papers

p 7 line 27-31, little attention is paid on whether practices learned may be sustained beyond the intervention. How will this be measured/quantified

We won't be measuring this in this trial within this study due the time scale of the rct. We are following them up at 6 months from baseline.

Why does the trial need 3 arms? Is this perhaps overcomplicated? We have two intervention arm (f2f and online) and one usual care arm (control). Pla's in the past have always been done f2f and have proved to be successful in LMIC's. We introduced the online arm as this a HIC and post pandemic there has been more inclination towards online meetings than f2f. We want to assess the feasibility of running sessions online, their efficacy compared to f2fa and the feasibility to train CFs for running both online and f2f sessions.

Hybrid model (online) - recognising how the communities have changed in ways they interact with systems, considering urban busy lifestyles/young children. Because we are working with marginalized communities (lack of accessibility to tech - so fff arm had to be included).

Specific comments

p 6 lines 12-17, what are the disparities faced? Why not be clear and state numbers, direction Britain's ethnically diverse population is mostly disadvantaged across a range of socioeconomic outcomes, forming fundamental causes of ethnic health inequalities in the UK [3]. Some of the widest differences have been observed in the South Asian (SA) population, more so for Pakistani and Bangladeshi communities compared to Indians rendering them more prone to limiting long term illnesses and poor health compared to the White British population[4]. These can mostly be attributed to but not limited to their disparities in socioeconomic class, health service access, language barriers leading to lack of knowledge and discrimination based on ethnicities [3].

p 6 lines 27-40, this paragraph is written as if there is a deficit in what the south asian community members are doing. Please consider reframing using an assets based framework
This section was revised: Effective early life interventions tailored to different ethnic groups have great potential to reduce the development of short- and long-term conditions and, thereby, lifetime inequalities. At present, however, few of such interventions exist [8,9]. Traditionally, UK health services provide unidirectional information based on guidelines and NHS recommendations. Specific ethnic groups may be marginalized by this approach as most advice is not tailored to their cultural practices. Hence, there is a need for interventions that not only target these communities through a top down unidirectional approach but instead bring a change that originates from within these communities. To do this it is important to work in partnership with these communities to help build their capacity and empower them to work closely with local authorities and stakeholders. The asset based community development approach also known as the ABCD approach has been successful in many areas for community development as this helps communities identify their assets through asset mapping and mobilize them to bring along the desired change.

p 7 line 31, many more details on the 'online' arm are needed. Who will access the information? will it be acceptable if a husband brother has the computer or cell phone? Information in the online arm will be accessible to the participant attending. Whilst we have no way of monitoring who is listening to the sessions we could expect anyone in the background to be able to listen to information being discussed. All participants have signed consent forms and understand the importance of confidentiality. The CFs would not collect any confidential identifiable information in the sessions (eg. child name, address, age etc).

Additionally through the online arm we are trying to assess the feasibility and acceptability of running these sessions online for the definitive trial by observing and collecting participant feedback on:

If people felt uncomfortable with online sessions?

Did they feel intruded?

How or did it impact their privacy, confidentiality?

p 8, what was codesigned? How do you think the co design influenced the intervention method

The research has been co-designed with SA CFs involved in all stages of developing and evaluating the intervention to ensure relevant research questions and an acceptable study design.

In the first phase of the NEON study which involved intervention development the CFs worked collaboratively to develop a culturally sensitive NEON intervention package consisting of (1) PLA group facilitator manual, (2) picture cards detailing recommended and non recommended feeding, care and dental hygiene practices with facilitators/barriers to uptake as well as solutions to address these, (3) healthy infant cultural recipes, (4) participatory Community Asset Maps and (5) list of resources and services supporting infant feeding, care and dental hygiene practices.

This phase of the NEON programme demonstrated the value of a collaborative approach between the target population, facilitators from the community and the researchers when developing a public health intervention. Such interventions that are co developed with members of the community help in recognising both social and cultural norms that may be of particular value for infants from ethnically diverse communities resulting in being not only more acceptable for the target population but also effective.

The community members who worked as facilitators in the intervention development phase, representing different SA communities, are now recruited as Community Researchers (CRs) to support;

- protocol development and ethics application by ensuring participants-facing documents are clear, appropriate and sensitively-worded;
- topic guide and questionnaire development;
- strategy development and troubleshooting (e.g. recruitment);
- interpretation of findings into appropriate and attainable recommendations for practice;
- review and revision of draft academic papers;
- dissemination activities and development of plain language summaries.

p 8 lines 41-44 need more details on the randomization and the way randomization would be concealed

Randomization of wards will be done before the recruitment by a separate member at UCL (not the RA) using the randomiser software. The randomised wards will be shared with the RA at the end of recruitment. Recruitment will be performed by health visitors largely, midwives, gps, community researchers who will not be aware of the randomisation. Once recruitment will be completed, the RA will assign participants to their respective arms. Baseline measurement of BMI will be done by health visitors who will be blinded. CFs/CRs will be made aware of the participant allocation before the PLA sessions start to make contact and invite participants.

p 9 lines 11-24, how will CFs be paid? trained? if trained, by whom? what topics?

A comprehensive manual for the CFs will be developed. The manual will be covering various areas/topics around nutrition with Women and Children first. Other resources (tool kit) will be provided to support with the learnings. The CFs will attend a three day comprehensive workshop to train on running sessions and one session will be conducted in between the 8 meeting cycle as a refresher and address any challenges faced. Biweekly meetings with the RA will be conducted to discuss learnings from the sessions. The CFs will be paid through vouchers for their time similar to a zero hour contract.

p 10, lines 20-35, recruitment through CBOs? Will information be retained on how the participant was recruited? Could the method of recruitment bias the sample? We have used multiple channels to support the recruitment - health visitors, midwives, gps, community researcher/facilitators, online advertisements on community social media pages. To ensure inclusivity, these methods are drawn from previous experiences from the research team. (recruitment is blinded to allocation, recruiters don't know, because this is an rct allocation to the arms is not dependent on recruitment

p 11, line 58 what does " CF unblinded debrief mean?"

Cfs are not involved in randomisation. We agree that this needs to be removed and is creating confusion. We have removed this.

In summary, if these points are considered by the authors, I believe this study protocol will make a nice contribution to the field. Thank you for the opportunity to comment.

Reviewer: 3

Dr. Jessica Kerr, University of Otago Christchurch

Comments to the Author:

Abstract & Intro

- NEON in full. Done

- Sometimes there are no spaces in between the brackets, please add the spaces where appropriate. Done

- Typo page 4, line 16 - Done

- The primary and secondary outcomes are unclear, which is BMI z-score? I think BMI is the proposed primary for the full trial, but not for the pilot? Please make that clear in abstract.

This was revised, the pilot will only assess for participants reporting on BMI z-score

- Unclear sentence page 6, line 38

Thank you, this was revised

- I felt like there was too many acronyms to keep track of and it made reading and following difficult. We have made some revisions, however due to word count constraints we were not able to reduce further.

Methods

- Because it's expected to run from July 2021, it was unclear what stage this was at? Perhaps the authors could now update with more accurate dates or estimates? We are currently in the intervention delivery phase. We completed recruitment in September 2022 and started with the PLA sessions in Oct 2022. We will finish the 8 meetings PLA cycle (held biweekly) in February 2023. The dates were pushed this far mainly due to the pandemic resulting in staff shortages within NHS, other pressing priorities and the Research Fellow being replaced.

- I also wondered if the protocol may need to be updated to clarify how the intervention was impacted by covid restrictions and how these intricacies will be handled in roll out and analysis etc

Thank you, In addition to the INTRODUCTION section where we describe the phase of the study this manuscript refers to : This protocol describes NEON Phase 2: Pilot Feasibility Cluster RCT.

we wanted to include the following section however, this will raise word count,
CURRENT STUDY STATUS (AS OF JAN 2023)

The study is currently in the intervention delivery phase.

There has been an overall delay in the timeline of the study due to the COVID - 19 pandemic, lockdown and NHS staff shortage in our study wards.

Recruitment for the study commenced in March 2022 and was completed in Sep 2022. Following recruitment, we launched our first PLA meeting in Tower Hamlets in Oct 2022 and then in Newham in the week following. Waltham Forest dropped out of the study due to lack of supporting staff in the recruitment phase.

We have completed 5 out of the 8 PLA meetings and the PLA meeting cycle will conclude in Feb 2023. Following this we will initiate data analysis and reporting of study findings. We will collect follow-up data in April 2023 which will be the last data collection time point for this trial.

- Again, I think there are too many acronyms and abbreviations, I don't think these are all needed We have made some revisions, however due to word count constraints we were not able to reduce further..

- What is the rationale for excluding mothers < 18 years?

Minors would not be included to maintain inclusivity and confidentiality amongst participants. In addition to that, to protect the young minors from social stigma.

- Please revise tense throughout methods, sometimes these swap between past, present and future tense.

This was revised, thank you!

- What training will facilitators receive to ensure this is standardised? Women and Children First are an organization who have worked on PLA's in LMICs and have expertise and experience of delivering such interventions. They have been involved in developing and finalizing our NEON toolkit and have trained the Community facilitators. We had a three day comprehensive workshop for the CFs and an additional session midway during the 8 meeting cycle.

- Outcome section table is nice, but perhaps confusing about which are the primary outcomes for pilot vs. full trial and as this paper is a protocol for the pilot, not the trial. Perhaps some colour coding or an extra column could be used to clarify this.

First table is for definitive trial however table 2 refers to the pilot RCT

- What is the rationale for only 1 of the outcomes measured at 12 months (partners time questionnaire) - According to the original timeline our 6 month data collection time-point is in Jan 2023. Post that in the next months until May 2023 (when the study grant period concludes) we would have only worked on data analysis, evaluation and report writing in collaboration with our respective partners from Newham and TH to assist/advise on data analysis. Hence, the partners time questionnaire (which is their time log) was planned to be collected until May 2023 as this would give us the total cost of implementation. All other outcomes are to do with participants and CFs who would finish off earlier (Jan 2023).

- Outcome table – please clarify if the 6m and 12m measures are after baseline or after study end? Partner's time? Partner of who - all outcomes will be measured at baseline, 14 weeks (end of PLA sessions) and 6 months from baseline. Partners are our study partners from Newham and Tower hamlets involved in the recruitment and delivery of NEON.

- The subheadings used in outcomes section are at odds with those listed in the abstract, please make sure it is always clear what the primary/secondary outcomes are for the pilot vs. proposed full trial. Done

- Will BMI z-score be converted to determine overweight/obesity, or only retained as continuous measure?

The BMI score will be recorded as a continuous measure. For the purpose of this pilot study we want to assess the feasibility of collecting the BMI. We will also categorise into over weight/obesity,

Analysis

- As analysis will be done during the intervention, who will do this and will they be/remain blinded? Statistical data analysis will be done by a statistician who would be blinded. outcome data analysts will be blinded to the identity of the study arms at both interim and final analyses. The steering committee will also be blinded to the identity of the study arms during any interim analyses.

- Will intention to treat analysis be considered?

Analysis section was revised to include: An intention to treat analysis will also be considered for completeness.

- Will there be sensitivity analysis for facilitators or assessors unblinded during the intervention/measurement periods?

RA will be blinded - this won't introduce any bias. CFs cannot be blinded due to the design of the study

For both quant and qual - no risk of bias

Separate the participants feedback form from the main analysis to avoid bias.

sensitivity approaches documented included the impact of noncompliance or protocol deviations, the impact of missing data, the impact of competing risks in a trial, the impact of a baseline imbalance, and the impact of different assumptions underlying the statistical model

We will use mixed-methods analysis to understand intervention and trial feasibility, acceptability and fidelity to participants and community facilitators alongside assessing equity impact and informing design of the definitive trial.

Quantitative data analysis will involve descriptive summary measures with expressions of uncertainty using the 95% confidence interval, including the parameters for sample size calculation for the definitive trial. For the primary outcome measure of the definitive trial, child BMI z-score, we will additionally estimate the intervention effect with confidence intervals between each of the intervention arms versus the control arms.

Quantitative findings will be complemented by a qualitative thematic framework analysis of the intervention implementation processes.

Quantitative data will be descriptively summarised using mean and standard deviation (SD) for continuous variables, number with percentages for categorical variables, and 95% CI with a breakdown of participants by trial arm, ethnic/language group and borough where relevant.

References

Could do with some reformatting and checking - edited

Reviewer: 1

Competing interests of Reviewer: None

Reviewer: 2

Competing interests of Reviewer: I have no competing interests and I understand the above statement and consent to the named publication of my review.

Reviewer: 3

Competing interests of Reviewer: No competing interests